# Curcumin in the Treatment of Kidney Disease: A Systematic Review with a Focus on Drug Interactions

**DOI:** 10.3390/antiox14111369

**Published:** 2025-11-18

**Authors:** Ebenezer Ofori-Attah, Abigail Aning, Layla Simón

**Affiliations:** 1Faculty of Pharmaceutical Sciences, Sojo University, Kumamoto 860-0082, Japan; g2271d02@m.sojo-u.ac.jp; 2Department of Clinical Pathology, Noguchi Memorial Institute for Medical Research, University of Ghana, Accra P.O. Box LG 581, Ghana; aaning@noguchi.ug.edu.gh; 3Escuela de Nutrición y Dietética, Universidad Finis Terrae, Santiago 7501015, Chile

**Keywords:** *Curcuma*, renal disease, diabetic nephropathy, inflammation, oxidative stress, fibrosis

## Abstract

Kidney disease (KD) is a major health challenge, affecting millions of people worldwide, highlighting the need for improved prevention and management strategies. The pathophysiological mechanisms converged on a common pathway characterized by inflammation, oxidative stress, fibrosis, nephron loss and failure. Curcumin, the active compound derived from turmeric (*Curcuma longa*), attracts considerable interest as a potential therapy for KD due to its anti-inflammatory, antioxidant and anti-fibrotic properties. Despite the benefits of curcumin, co-administration with kidney medications may cause drug interactions. Here, we systematically reviewed the efficacy of curcumin in alleviating KD and its safety when used with conventional treatments. Search terms included: curcumin AND (“diabetic nephropathy” OR “renal disease” OR “kidney disease”). Data on mechanisms of action, redox status, clinical benefits, side effects, and drug interactions were extracted and analyzed. Curcumin reduces oxidative stress, inflammation, apoptosis, fibrosis, ER stress, and lipid and glucose metabolism. Curcumin has multifaceted nephroprotective effects, while it is safe and well-tolerated. The curcumin–drug interactions reviewed were: -piperine, -epigallocatechin gallate, -losartan, -ginkgolide B, -rosuvastatin, -insulin, -cilostazol, and -ginger. These interactions improve curcumin bioavailability, and synergistic anti-inflammatory/antioxidant/antifibrotic and renoprotective effects. Future research should prioritize large-scale clinical trials to evaluate the efficacy and safety of curcumin in diverse KD populations.

## 1. Introduction

Kidney disease is a major global health challenge, affecting hundreds of millions of people worldwide [1,2,3]. Chronic kidney disease (CKD) and acute kidney injury (AKI) are the most prominent forms, each associated with high morbidity, mortality and economic burden [4,5,6]. Globally, CKD affects approximately 9–13% of the population, with diabetes mellitus and hypertension being the main causes [7,8]. Acute kidney injury, which is common in hospitalized and critically ill patients, can further accelerate the development of CKD. The prevalence of kidney disease varies according to region, socio-economic status and access to healthcare, with low- and middle-income countries bearing a disproportionate burden [1,9,10]. With rising rates of diabetes, obesity and ageing populations, the global impact of kidney disease is expected to increase [8], highlighting the urgent need for improved prevention and management, and early detection. Treatment of kidney disease depends on the underlying cause. It usually involves slowing the progression of the disease and treating symptoms or complications with medications including angiotensin-converting enzyme (ACE) [11], or angiotensin receptor blockers (ARBs) [12], sodium-glucose transport proteins 2 (SGLT2) inhibitors [13], fluid and electrolyte balance [14] and other supportive medications.

Kidney disease, whether acute or chronic, results from a variety of initiating factors. Despite these diverse origins, such as metabolic dysfunction, immune attack, toxic exposure or vascular injury, the pathophysiological mechanisms involved tend to converge on a common pathway characterized by inflammation, fibrosis, nephron loss and eventual functional failure [15,16,17,18]. Following injury, a complex inflammatory response is triggered. This inflammatory environment not only exacerbates cellular injury but also promotes the generation of reactive oxygen species (ROS), leading to further oxidative damage to renal cells [19,20,21]. Therefore, reducing oxidative stress is emphasized as a useful way to treat kidney disease.

Curcumin, the active compound derived from turmeric (*Curcuma longa*), has attracted considerable interest in the scientific community due to its anti-inflammatory, antioxidant and anti-fibrotic properties [22,23,24,25]. These properties make curcumin a potential therapeutic agent for kidney disease, which is often driven by chronic inflammation, oxidative stress and fibrosis. Curcumin inhibits nuclear factor-kappa B (NF-κB), a key regulator of inflammation [26], and transforming growth factor-beta (TGF-β), a key cytokine that promotes fibrosis in kidney tissue [27]. Curcumin neutralizes ROS and enhances the activity of endogenous antioxidants such as superoxide dismutase (SOD) and glutathione peroxidase [28,29]. Curcumin has been shown to reduce serum creatinine, blood urea nitrogen (BUN) and urinary albumin/protein in animal models of renal injury [30,31,32,33]. Animal studies suggest protective effects against ischemia–reperfusion injury and drug-induced nephrotoxicity [34].

Research on natural remedies, particularly curcumin for prevention and treatment of renal disease, has gained popularity. However, clinical evidence and potential interactions with medications used to treat kidney disease are limited. Curcumin inhibits several cytochrome P450 enzymes (e.g., CYP3A4, CYP2C9) [35,36,37], which play relevant roles in the metabolism of many drugs, including those used to treat kidney disease [38,39,40]. If curcumin modulates these enzymes, it may increase the plasma levels of the drugs being metabolized, potentially leading to toxicity. Therefore, understanding these drug interactions is crucial, especially for individuals who take multiple medications for chronic conditions [41].

Despite the benefits of curcumin, co-administration with kidney medications may cause drug interactions. Here we reviewed the efficacy of curcumin in alleviating kidney disease and its safety when used with conventional medications.

## 2. Methods

### 2.1. Study Design

This systematic review followed the Preferred Reporting Items for Systematic Reviews and Meta-Analyses (PRISMA) guidelines and was registered in the Open Science Framework (https://osf.io/kvxf2, registered on 15 May 2025).

### 2.2. Search Strategy

The systematic review was conducted in April 2025 using PubMed, Scopus, clinicaltrials.gov, and Google Patents as search databases. Two researchers independently extracted data to ensure accuracy and reliability. The research question was: What are the drug interactions with curcumin in treatments of patients with kidney disease? This question was structured using the PICOS framework, where P (Population) corresponds to patients with kidney disease, I (Intervention) refer to a curcumin treatment, C (Comparison) represents a treatment without curcumin, O (Outcome) focuses on the effects for kidney disease, and S (Setting) specifies the context of curcumin and drug interaction in kidney disease. Keywords and search terms included: curcumin AND (“diabetic nephropathy” OR “renal disease” OR “kidney disease”).

### 2.3. Analysis of Scope and Feasibility

Our systematic review summarized the evidence published between 2021 and 2025 about the effects of curcumin on kidney disease. In this way, we actualized the information systematically reviewed previously [42,43,44] with a novel focus on drug interaction.

### 2.4. Selection Criteria

Table 1 summarizes the inclusion and exclusion criteria. This study analyzed original articles published between 2021 and 2025 in English; focus on models of kidney disease; drug interactions; and interventions involving humans and animals. On the other hand, the exclusion criteria for selecting research articles were articles in languages different to English; studies that did not report original data; studies for pathologies different to renal disease; studies with curcumin as a single treatment; and only in vitro analysis.

### 2.5. Data Analysis

Data on mechanisms of action, redox status, clinical benefits, side effects, and drug interactions were extracted and analyzed. Excel software (version 2509) was employed to organize and manage the selection process, including identifying original articles, duplicate entries and screening articles based on their titles and abstracts (Appendix A).

## 3. Results

A total of 742 articles were identified during the initial search process. In the first screening phase, 263 non-original articles were removed. Then, 48 duplicate articles were removed. Subsequently, in the last screening phase, an evaluation of titles and abstracts excluded an additional 412 articles. Ultimately, 19 original articles met the inclusion criteria and were used for the final analysis (Figure 1).

The characteristics of the selected articles are summarized in Table 2. Briefly, one article reported in vitro and in vivo effects of curcumin; twelve studies were developed in animals; and six were performed in humans.

## 4. Discussion

Table 2 summarizes the principal findings of each article. Briefly, curcumin alone or in combination inhibits EGFR/ERK1/2, JNK, PI3K/mTOR, and p38 MAPK pathways in MDCK kidney cells (Figure 2) [45]. Moreover, curcumin exerts antioxidant, anti-inflammatory, and antifibrotic properties and reduces kidney cyst size and number in mice [45,54]. In rats, curcumin replicates effects observed in mice, including reductions in serum glucose, lipid, creatinine and urea levels, and reverses proteinuria, while protecting and improving renal function [46,47,48,49,50,51,52,53,54,55,56,57]. In human studies, curcumin is safe and well-tolerated, reduces symptoms associated with kidney disease, and prevents and preserves renal function [58,59,60,61,62,63].

### 4.1. Mechanisms of Actions of Curcumin in Kidney Disease

Curcumin exerts pleiotropic actions targeting oxidative stress, inflammation, apoptosis, endoplasmic reticulum (ER) stress, fibrosis, and metabolic dysregulation [64,65] (Table 2 and Figure 3). Its primary renoprotective mechanism involves the attenuation of oxidative stress, a key driver of chronic kidney injury. Curcumin enhances the activity of intrinsic antioxidant enzymes such as superoxide dismutase (SOD) and catalase (CAT), while decreasing levels of malondialdehyde (MDA), a biomarker of lipid peroxidation. This antioxidant activity is crucial for neutralizing ROS and preventing ROS-induced cellular damage in renal tissues [47,49,53].

Additionally, curcumin exerts anti-inflammatory effects by downregulating nuclear factor-kappa B (NF-κB) and its downstream pro-inflammatory cytokines, including interleukin-6 (IL-6), tumor necrosis factor-alpha (TNF-α), and C-reactive protein (CRP). Suppression of these inflammatory mediators helps to interrupt the cycle of inflammation-mediated renal injury and fibrosis. Curcumin also modulates apoptotic signaling pathways. Curcumin reduces the expression of pro-apoptotic markers Bax and caspase-3 while upregulating Bcl-2, an anti-apoptotic protein, thereby preserving tubular epithelial cell viability. This anti-apoptotic action may be critical in mitigating tubular atrophy, a hallmark of CKD [50].

Furthermore, curcumin downregulates the expression of fibrogenic markers such as transforming growth factor-beta 1 (TGF-β1), p38 mitogen-activated protein kinase (p38 MAPK), c-Jun N-terminal kinase (JNK), and the mammalian target of rapamycin (mTOR). These signaling pathways are central to the development of tubulointerstitial fibrosis, and their inhibition by curcumin contributes to structural preservation of renal tissue. In addition, curcumin alleviates endoplasmic reticulum (ER) stress, as evidenced by decreased expression of ER stress markers such as C/EBP homologous protein (CHOP) and glucose-regulated protein 78 (GRP78). By restoring ER homeostasis, curcumin may protect against misfolded protein accumulation and ER-mediated apoptosis [47,53].

In terms of metabolic regulation, curcumin significantly reduces circulating levels of glucose, low-density lipoprotein (LDL), and triglycerides, suggesting an ameliorative effect on diabetic nephropathy and metabolic syndrome-associated kidney injury. These effects are particularly relevant given the high prevalence of metabolic disorders among patients with CKD [51,53,57,62].

### 4.2. Clinical Benefits and Adverse Effects of Curcumin in Kidney Disease

Emerging experimental and clinical evidence highlights curcumin’s multifaceted nephroprotective effects, including its antioxidant, anti-inflammatory, antifibrotic, and metabolic regulatory properties. These pharmacodynamic actions are especially relevant in the pathophysiology of CKD, diabetic nephropathy, and AKI, where oxidative stress and inflammation serve as pivotal drivers of progression. Additionally, curcumin reduces kidney damage, improves renal function, is safe and well tolerated [45,46,47,48,49,50,51,52,53,54,55,56,57,58,59,60,61,62,63,64]. In a diabetic nephropathy rat model, a patented traditional Chinese medicine containing curcumin, reports beneficial effects reducing serum creatinine and urea nitrogen [57]. Another patented pharmaceutical composition containing curcumin reduces proteinuria in diabetic nephrotic patients [63]. Clinically, curcumin has demonstrated beneficial outcomes in randomized trials, such as reductions in proteinuria, serum creatinine, and inflammatory biomarkers. For instance, in lupus nephritis patients, curcumin supplementation led to significant improvements in hematuria and proteinuria, suggesting its immune-modulatory and anti-inflammatory capacity [66].

Despite these encouraging findings, curcumin’s clinical translation is hindered by its poor oral bioavailability, attributed to low solubility, rapid metabolism, and systemic elimination [67]. Innovations such as curcumin–piperine co-formulations, phospholipid complexes, and nanoparticle-based delivery systems have shown promise in overcoming these limitations, though standardized pharmacokinetic profiles remain elusive [65]. Although generally well-tolerated at low to moderate doses, high-dose curcumin can induce gastrointestinal discomfort, interfere with iron absorption, a concern in CKD-related anemia, and interact with drug-metabolizing enzymes [34,68]. The risk of adverse interactions, especially in polypharmacy common among CKD patients, underscores the need for comprehensive drug–supplement interaction studies.

### 4.3. Curcumin Interactions in Kidney Disease

#### 4.3.1. General Pharmacokinetic and Safety Interactions of Curcumin

In the clinical management of kidney disease and its associated comorbidities, patients are often prescribed complex polypharmacy regimens, which heighten the risk of adverse drug–drug interactions. Such interactions can alter the pharmacokinetics and pharmacodynamics of co-administered medications, potentially diminishing therapeutic efficacy, increasing toxicity, or exacerbating pre-existing conditions. Emerging evidence indicates that curcumin significantly interacts with several drug classes and metabolic pathways, notably cytochrome P450 (CYP) enzymes and P-glycoprotein (P-gp) transporters, which play crucial roles in drug metabolism and disposition [69,70].

#### 4.3.2. Modulation of CYP Enzymes and Drug Transporters (P-gp)

Curcumin and its metabolites interact with the drug-metabolizing machinery, particularly the CYP enzyme system and drug efflux transporters such as P-gp [69]. Curcuminoids can exert moderate to potent inhibitory effects on several CYP isoforms: CYP-2C19, -2B6, -2C9, -3A (Table 3) [71]. Notably, CYP3A4, a major enzyme responsible for metabolizing a vast array of therapeutic drugs, is significantly inhibited by curcumin [72]. This inhibition can lead to decreased metabolism of numerous co-administered drugs, potentially increasing their plasma concentrations and their pharmacological effects or adverse reactions. Piperine, frequently combined with curcumin to enhance its bioavailability, is a relatively selective non-competitive inhibitor of CYP3A, and also affects CYP1A1, 1B1, 1B2, and 2E1, further contributing to this interaction profile [72].

Regarding P-gp (such as ABCB1), curcumin acts as an inhibitor [73,74] and its chronic administration has been shown to down-regulate intestinal P-gp levels, which can lead to increased absorption of P-gp substrate drugs (Table 3) [75]. Conversely, hepatic P-gp levels may be upregulated, while renal P-gp levels generally remain unaffected [69]. These complex, tissue-specific modulations of CYP enzymes and P-gp can significantly alter the pharmacokinetics of co-administered medications, potentially leading to unpredictable changes in drug efficacy or toxicity.

#### 4.3.3. Effects on Coagulation and Platelet Function

Turmeric and its active constituent, curcumin, have demonstrated inherent antiplatelet and anticoagulant properties. In vitro studies have shown that curcumin can inhibit arachidonic acid-induced platelet aggregation [3]. Turmeric may increase the risk of bleeding by interfering with the clotting cascade, specifically by decreasing platelet aggregation. Clinical case reports have documented instances of increased International Normalised Ratio (INR), a measure of blood clotting time, in patients taking warfarin after initiating turmeric-containing products [76]. Increased clopidogrel levels were also observed in an animal study with high-dose curcumin (Table 3) [77]. On the other hand, curcumin increase the risk or severity of thrombosis when combined with erythropoiesis-stimulating agents (ESAs) such as Darbepoetin alfa, Erythropoietin, or Peginesatide [78]. This highlights the need for clinicians to be aware of both general anticoagulant properties and specific drug–drug interaction alerts.

#### 4.3.4. Impact on Blood Glucose Regulation

Studies have indicated that curcumin can decrease blood glucose and glycosylated hemoglobin (HbA1c) levels in diabetic rats and patients (Table 3) [57,79]. One pharmacokinetic study demonstrated that curcumin increased glyburide (a conventional antidiabetic drug) blood levels and significantly decreased plasma glucose levels for up to 24 h, although no overt hypoglycemia was reported [80]. This interaction necessitates rigorous monitoring of blood glucose levels in diabetic kidney disease patients who are taking curcumin supplements and may require adjustments to their conventional antidiabetic drug dosages.

**Table 3 antioxidants-14-01369-t003:** General Pharmacokinetic and Safety Interactions of Curcumin.

Interaction Category	Mechanism/Effect	Clinical Implication/Risk	Ref.
**Cytochrome P450 (CYP) Enzymes**	Curcuminoids inhibit various CYP isoforms (CYP2C19, CYP2B6, CYP2C9, CYP3A, CYP1A2, CYP2D6).	Decreased metabolism of co-administered drugs, leading to increased systemic exposure, higher plasma concentrations, and potential for increased pharmacological effects or adverse reactions. Complex and tissue-specific modulation requires careful monitoring.	[71]
**P-glycoprotein (P-gp)**	Curcumin inhibits P-gp (efflux pump). Chronic curcumin administration can down-regulate intestinal P-gp, but up-regulate hepatic P-gp.	Altered absorption and distribution of P-gp substrate drugs, potentially increasing their bioavailability and leading to unpredictable changes in efficacy or toxicity.	[74,81]
**Coagulation/Platelet Function**	Curcumin has antiplatelet effects (inhibits platelet aggregation, interferes with clotting).	Increased risk of bleeding and bruising, especially when co-administered with anticoagulants (e.g., warfarin, clopidogrel) or antiplatelet agents.	[77,82]
**Blood Glucose Regulation**	Curcumin can lower blood glucose and HbA1c levels.	Reduce glucose level when co-administered with antidiabetic drugs (e.g., glyburide). Requires rigorous blood glucose monitoring.	[57,79]

### 4.4. Systematic Analysis of Curcumin-Drug Combinations in Kidney Disease

A significant obstacle to curcumin’s therapeutic efficacy is its inherently poor oral bioavailability, which is characterized by inadequate absorption from the gastrointestinal tract, rapid systemic metabolism, and swift elimination from the body [83]. When curcumin is administered orally in its unformulated form, it typically results in very low or even undetectable plasma concentrations. For instance, a study in rats revealed that only trace amounts of the compound were present in urine following oral administration, with approximately 75% of the dose excreted in feces [67]. This poor bioavailability demands the use of advanced strategies to achieve therapeutically relevant systemic concentrations.

One such approach involves the co-administration of piperine, an alkaloid derived from black pepper, which is a well-established enhancer of curcumin absorption. In human volunteers, piperine has been shown to increase curcumin’s bioavailability by as much as 2000% [84]. It functions by inhibiting CYP450 isoforms and drug efflux transporters such as P-gp in both the gut and liver, thereby slowing the breakdown and excretion of curcumin [85].

Another effective strategy is the use of nanoformulations, including polymeric nanoparticles, copolymeric systems, nanocrystals, and nanovesicles. These nanocarrier systems significantly improve curcumin’s bioavailability and therapeutic potential by increasing its concentration in target tissues. For example, an EGCG-based nano-antioxidant formulation (Cur@EK) has demonstrated markedly improved oral and systemic bioavailability compared to free curcumin. Similarly, nanocurcumin (PLGA-GA2-CUR) has been specifically designed to overcome the limitations associated with conventional curcumin delivery [54,86].

Additionally, lipid-based delivery systems have shown promise in enhancing curcumin’s absorption and systemic availability. Formulations using phospholipid carriers, such as polyenylphosphatidylcholine (PPC), facilitate the efficient transport of curcumin and its metabolites across the intestinal barrier. This approach has been particularly effective in improving the bioavailability of active metabolites like tetrahydrocurcumin (THC), thereby optimizing the compound’s therapeutic benefits [55].

Eight studies within this systematic review demonstrated that curcumin-containing products and other drugs exert significant interactions in CKD (Table 4).

#### 4.4.1. Turmeric and Piperine

As described above, piperine is an alkaloid extracted from black pepper (*Piper nigrum*), primarily recognized as a bioavailability enhancer and widely utilized to improve curcumin’s poor bioavailability [87]. Piperine dramatically increases curcumin’s absorption by as much as 2000% [84]. This enhancement is achieved through several mechanisms, including the inhibition of various drug metabolism enzymes (e.g., CYP1A1, -1B1, -1B2, -2E1, -3A4) and P-gp, as well as by stimulating amino acid transporters and decreasing glucuronic acid production. Enhanced bioavailability means higher systemic exposure to curcumin, which can potentially amplify its anti-inflammatory and antioxidant effects, thereby improving its therapeutic impact [87]. A randomized double-blind clinical trial demonstrated that curcumin and piperine combination is more effective than turmeric alone in attenuating oxidative stress and inflammation in hemodialysis patients [61].

Piperine itself can increase the bioavailability of various drugs (e.g., barbiturates, dapsone, ethambutol, isoniazid, phenytoin, propranolol, rifampicin, sulfadiazine, theophylline, fexofenadine) by affecting CYP450 isoenzymes [87]. While the turmeric and piperine combination is highly effective at overcoming curcumin’s bioavailability limitations, it simultaneously magnifies the potential for adverse drug interactions. This poses a significant challenge for safe clinical integration and necessitates rigorous monitoring, especially in patients with kidney disease who are often on complex medication regimens.

#### 4.4.2. Nanoparticle-Encapsulated Curcumin and Epigallocatechin Gallate (EGCG)

Epigallocatechin Gallate (EGCG) is a polyphenol, specifically a catechin, abundantly found in green tea [86]. A novel EGCG-based nano-antioxidant (EK) has been developed to enhance the bioavailability and anti-inflammatory efficacy of curcumin (Cur@EK), particularly in conditions such as acute colon and kidney inflammation [86].

EK nanoparticles exhibit radical-scavenging capabilities and regulate redox processes within macrophages in vitro, which are key components in the inflammatory response. Curcumin-EK demonstrates a synergistic anti-inflammatory effect, reducing the levels of pro-inflammatory cytokines such as TNF-α and IL-6, while increasing the anti-inflammatory cytokine IL-10 in lipopolysaccharide-stimulated macrophages [86]. In mouse models of AKI, Cur@EK shows pronounced anti-inflammatory effects. It leads to a significant reduction in blood urea nitrogen (BUN) and serum creatinine (CRE) levels, critical markers for assessing kidney function, and results in minimal tubule damage observed on histology [86]. Nanoparticle-encapsulated curcumin and epigallocatechin gallate reduce streptozotocin-induced diabetic nephropathy in mice. Curcumin exerts antioxidant (↓ MDA, ↑ SOD and CAT), anti-inflammatory (↓ TNF-α and IL-6), and antifibrotic properties [54].

The use of EGCG as a component of the nanoparticle itself to encapsulate curcumin, rather than just a co-administered agent, represents a sophisticated approach to drug delivery and synergistic action. This strategy leverages nanotechnology to overcome curcumin bioavailability issues and achieve synergistic anti-inflammatory and antioxidant effects that further support its therapeutic potential for future clinical applications.

#### 4.4.3. Tetrahydrocurcumin, Polyenylphosphatidylcholine, and Losartan

Tetrahydrocurcumin (THC) is the principal metabolite of curcumin, retaining significant antioxidant properties. Polyenylphosphatidylcholine (PPC) is a lipid carrier, often used to enhance the absorption and bioavailability of lipophilic compounds such as THC, which has low water solubility and poor enteric absorption. Losartan is an Angiotensin II Receptor Blocker (ARB), a class of antihypertensive drugs widely used in kidney disease management to reduce proteinuria and slow disease progression [45,55].

The triple combination of THC, PPC, and losartan demonstrated significant benefits in improving kidney outcomes in a rat model of type 2 diabetic nephropathy. The THC + PPC + losartan treatment significantly lowered blood pressure, a critical factor in managing diabetic nephropathy and a primary target of ARBs. The combined therapy led to increased levels of antioxidant copper-zinc-superoxide dismutase (CuZnSOD) in the kidneys, indicating an improvement in the defense against oxidative stress. The treatment significantly decreased the levels of protein kinase C-α (PRKCA), kidney injury molecule-1 (KIM-1), and type I collagen in the kidneys, suggesting reduced kidney damage and fibrosis. In fact, fibrosis was significantly attenuated on kidney histology, and tubulointerstitial injury was decreased. There was a trend towards decreased albuminuria and improved creatinine clearance in the THC + PPC + losartan group [55].

These beneficial changes in renal parameters were observed independently of glycemic status, as elevated plasma fructosamine levels were similar across the CKD groups [55]. This reinforces the potential for disease modification beyond glucose control, echoing findings with curcumin and insulin. Moreover, losartan is known to be activated by CYP3A4, suggesting a potential altered pharmacokinetics for losartan, requiring clinical monitoring.

#### 4.4.4. Curcumin & Ginkgolide B

Ginkgolide B (GB) is a terpene lactone derived from the leaves of Ginkgo biloba. It functions primarily as a Platelet-Activating Factor Receptor (PAFR) antagonist. Beyond this, GB possesses a range of biological activities, including antiplatelet, anti-inflammatory, antioxidant, and neuroprotective properties [89,90,91]. GB is known to interact with antiplatelet medications and anticoagulants, thereby increasing the risk of bleeding and abnormal coagulation. Mechanistically, GB inhibits platelet release by blocking Syk and p38 MAPK phosphorylation in thrombin-stimulated platelets, leading to reduced ATP release and decreased expression of platelet factor 4 (PF4) and CD40 Ligand (CD40L) [82].

The combination of curcumin and GB has shown promising synergistic effects, particularly in the context of Autosomal Dominant Polycystic Kidney Disease (ADPKD). Studies, both in vitro (using the Madin-Darby Canine Kidney (MDCK) cyst model) and in vivo (in a Pkd1 knockout mouse model), have demonstrated that curcumin and GB synergistically inhibit cyst formation and enlargement, which are hallmarks of ADPKD [45]. This is a significant finding given the limited effective treatments for this progressive genetic kidney disorder.

The observed synergistic effect is attributed to their combined ability to regulate multiple intracellular signaling pathways implicated in cystogenesis. Curcumin may block pathways such as EGFR/ERK1/2, JNK, and PI3K/mTOR, while GB may inhibit cystogenesis via downregulation of EGFR/ERK1/2, JNK, and p38 signaling pathways [45]. Both compounds independently reduced renal cyst cell proliferation by downregulating the Ras/ERK1/2 signaling pathway.

The synergistic anti-cystogenic effects observed in ADPKD models are highly promising, suggesting a potential combination therapy for a disease with significant unmet clinical needs. However, the combined antiplatelet effects of both curcumin and GB raise significant safety concerns regarding bleeding risk.

#### 4.4.5. Curcumin & Rosuvastatin

Rosuvastatin is an HMG-CoA Reductase Inhibitor, commonly known as a statin. Statins are widely prescribed lipid-lowering agents used to manage dyslipidemia, a prevalent comorbidity in CKD patients that contributes significantly to cardiovascular risk [92]. It is noteworthy that rosuvastatin alone, even at a minimal therapeutic dose, impaired renal morphology in the rat study, despite showing beneficial effects on lipid profile [46]. A reduced dose of rosuvastatin alone, without curcumin, showed no treatment potential for CKD in this model.

Research has explored the combination of curcumin with rosuvastatin in the context of adenine-induced CKD and associated dyslipidemia in rats [46]. In this animal model, the combination of curcumin with a reduced dose of rosuvastatin demonstrated superior renal protection compared to CKD controls. This was evidenced by lower serum creatinine levels and milder renal morphological alterations, including reduced inflammation, interstitial fibrosis, and tubular degeneration. Concurrently, the combination achieved improved antilipemic action, with lower levels of triglycerides, very low-density lipoprotein (VLDL), and low-density lipoprotein (LDL) cholesterols [46].

A key aspect of this interaction is the pharmacokinetic influence of curcumin on rosuvastatin. Previous pharmacokinetic studies in rats and dogs have shown that co-administration of curcumin significantly increased rosuvastatin plasma concentration, area under the curve, and serum half-life [93]. This suggests that curcumin may inhibit rosuvastatin’s metabolism, potentially through its known inhibitory effects on certain CYP enzymes.

This combination demonstrates a promising strategy for managing both CKD progression and associated dyslipidemia. The interaction between curcumin and rosuvastatin exemplifies a beneficial pharmacokinetic interaction that allows for a reduced dose of a conventional drug, potentially mitigating its inherent side effects while maintaining or enhancing therapeutic efficacy.

Despite these promising preclinical results, larger animal trials and rigorous clinical trials are required to further investigate the therapeutic potential of this combination, ensure its clinical relevance and effectiveness, and establish appropriate therapeutic protocols for human patients with CKD and associated dyslipidemia.

#### 4.4.6. Nanocurcumin Combined with Insulin

Insulin is an antidiabetic hormone, critical for regulating glucose homeostasis in individuals with diabetes mellitus [94]. Diabetic kidney disease (DKD) is a severe long-term complication of both type 1 and type 2 diabetes [95].

Research has explored the therapeutic potential of nanocurcumin combined with insulin in alleviating DKD in streptozotocin-induced diabetic rat models [48]. A key mechanism involves the deactivation of hyperactivated P38 (MAPK) and P53 targets in both kidney and liver tissues [48].

These pathways are crucial mediators of inflammation, oxidative stress, and apoptosis in DKD. Notably, this protective mechanism was observed to be independent of hyperglycemia moderation. The combination treatment, and surprisingly nanocurcumin alone, normalized or significantly improved impaired liver and kidney functions, as indicated by parameters such as BUN, creatinine, albumin, globulin, bilirubin, and alkaline phosphatase. Kidney histopathological abnormalities characteristic of DKD, such as basement membrane thickening, mesangial matrix expansion, tubular atrophy, and podocyte cytoskeletal impairment, were significantly mitigated by nanocurcumin and nanocurcumin + insulin treatments [48].

The ability of nanocurcumin to alleviate DKD independently of glycemic status when combined with insulin signifies a crucial shift in therapeutic strategy. This moves from a purely glucose-centric management to a more holistic approach that directly targets the underlying inflammatory and apoptotic pathways driving kidney damage. This approach could lead to better long-term outcomes for diabetic patients by mitigating the chronic complications of hyperglycemia on vital organs.

#### 4.4.7. Curcumin-Tagged Cilostazol

Cilostazol is a Phosphodiesterase-3 inhibitor [96], primarily used as an antiplatelet agent and vasodilator for symptomatic treatment in patients with peripheral ischemia [97]. Its mechanism involves inhibiting cAMP degradation, increasing intracellular cAMP in platelets and blood vessels, which suppresses platelet aggregation and induces vasodilation [97]. Cilostazol has also been shown to reduce plasma triglycerides and increase HDL cholesterol concentrations. Preclinical research has explored the therapeutic potential of a curcumin-tagged cilostazol solid nanodispersion for the management of diabetic nephropathy in Wistar rat models [52]. Cilostazol is known to inhibit ROS, while curcumin is a well-established antioxidant and anti-inflammatory agent. The combined inhibitory actions of cilostazol and curcumin contribute to their observed reno- and pancreas-protective effects [52].

The development of a curcumin-tagged cilostazol nanodispersion represents a sophisticated strategy to combine the benefits of both agents and improve their delivery for diabetic nephropathy. However, the potential for increased bleeding risk due to additive antiplatelet effects from both curcumin and cilostazol, coupled with potential pharmacokinetic interactions (CYP inhibition) that could increase cilostazol exposure, requires careful clinical consideration.

#### 4.4.8. Ginger and Curcumin

Both ginger (Zingiber officinale) and turmeric (containing curcumin) are widely used herbal supplements due to their culinary, anti-inflammatory, antioxidant, antiplatelet, and hypoglycemic properties [98,99]. Ginger and curcumin are frequently consumed together, leading to a cumulative pharmacological profile, such as reducing inflammatory proteins (cytokines) and cells [100]. While neither ginger nor curcumin specifically targets kidney function, their general anti-inflammatory and antioxidant properties could offer broad protection to the kidneys, demonstrated by the lower levels of urea and creatinine in dialysis patients treated with ginger [101]. Ginger is also notably low in potassium, which can be beneficial for kidney disease patients who often need to restrict their potassium intake. Ginger exhibits antiplatelet effects by inhibiting cyclooxygenase-1 (COX-1) and the arachidonic acid metabolism cascade, leading to a reduction in thromboxane B2 and prostaglandin synthesis [102]. As previously discussed, curcumin also has intrinsic antiplatelet properties. Combining ginger and curcumin can therefore lead to an increased risk of bleeding, particularly when co-administered with conventional antiplatelet or anticoagulant medications. Both ginger and curcumin have been reported to reduce blood sugar levels. Combining them with antidiabetic drugs may increase the risk of hypoglycemia [88]. The traditional co-consumption of ginger and turmeric for anti-inflammatory and antioxidant benefits for kidney health carries a heightened risk of bleeding and hypoglycemia, which necessitates careful patient counseling and monitoring by healthcare professionals.

Ginger and curcumin administration can reduce streptozotocin-induced diabetic nephropathy in Sprague Dawley rats [56]. In this model, curcumin performs antioxidant (↓ MDA, but ↑ CAT and SOD), anti-inflammatory (↓ IL-6 and NF-κB), antihyperglycemic, and renal protective effects [56].

## 5. Limitations and Challenges

While curcumin offers promising therapeutic benefits in CKD, several critical limitations must be acknowledged. Its poor oral bioavailability due to both low aqueous solubility and rapid metabolic degradation significantly restricts systemic efficacy. Moreover, the absence of standardized dosing guidelines and the limited availability of high-quality clinical trials in CKD populations collectively hinder its evidence-based application. In addition, curcumin’s potential to interact with commonly prescribed nephrological medications through modulation of cytochrome P450 enzymes and drug transporters such as P-glycoprotein raises considerable safety concerns, particularly in polypharmacy settings. Furthermore, formulation-dependent variations complicate clinical use, as enhanced-bioavailability products, though more effective systemically, may also increase the risk of toxicity and drug interactions. Long-term safety data are also lacking, and no validated biomarkers currently exist to monitor therapeutic response or adverse effects, which further limits its clinical utility. Additionally, curcumin’s effects on the gut microbiome, as an important modulator of its bioavailability, antioxidant, and pharmacological effects, are not reported within the 19 articles systematically reviewed. Lastly, regulatory inconsistencies across markets, along with variability in product quality and labeling accuracy, pose additional practical challenges. Therefore, addressing these gaps through rigorous pharmacokinetic studies, formulation standardization, and large-scale clinical trials is essential to support the safe and effective integration of curcumin into CKD management.

## 6. Conclusions and Future Directions

Curcumin holds considerable therapeutic promise in the management of CKD due to its multifaceted biological activities, including anti-inflammatory, antioxidant, and anti-fibrotic effects. These mechanisms intersect with key pathogenic pathways in CKD, suggesting a potential role for curcumin as an adjunctive treatment. However, despite encouraging preclinical and limited clinical evidence, its clinical translation remains constrained by poor oral bioavailability and a complex pharmacokinetic profile.

The safety of curcumin supplementation in CKD patients warrants careful consideration, particularly due to its dose-dependent adverse effects and potential for drug interactions, especially with medications possessing narrow therapeutic windows or those metabolized by CYP enzymes and P-gp. The risk may be exacerbated with bioavailability-enhanced formulations, which, although more effective systemically, may increase the likelihood of toxicity and pharmacokinetic interactions.

The principal curcumin–drug interactions reviewed were: -piperine, and -epigallocatechin Gallate (EGCG), -losartan, -ginkgolide B, -rosuvastatin, -insulin, -cilostazol, and -ginger. These drug interactions may improve curcumin bioavailability, synergistic anti-inflammatory/antioxidant/antifibrotic and renoprotective effects.

However, future research should prioritize well-designed, large-scale clinical trials to evaluate curcumin’s efficacy and safety in diverse CKD populations, including long-term administration across different formulations and co-administered medications, while also investigating potential systemic effects mediated by the gut microbiome, which may influence bioavailability and the gut–kidney axis. Importantly, detailed human pharmacokinetic and pharmacodynamic studies are essential to elucidate curcumin’s interaction profile, particularly with enhanced formulations. Mechanistic studies examining the modulation of drug-metabolizing enzymes and transporters in renal and hepatic tissues will improve the prediction and management of drug interactions. Such efforts will be instrumental in guiding evidence-based recommendations and optimizing the safe integration of curcumin into nephrological practice.

## Figures and Tables

**Figure 1 antioxidants-14-01369-f001:**
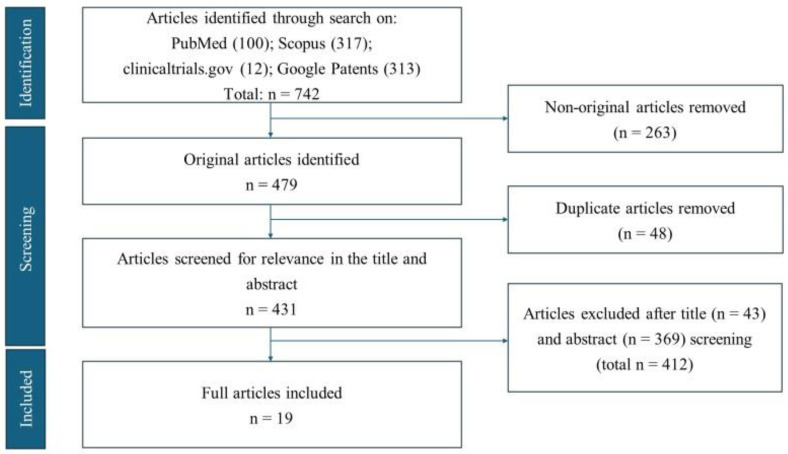
Flow diagram illustrating the identification and selection process of articles for the study, following the PRISMA guidelines.

**Figure 2 antioxidants-14-01369-f002:**
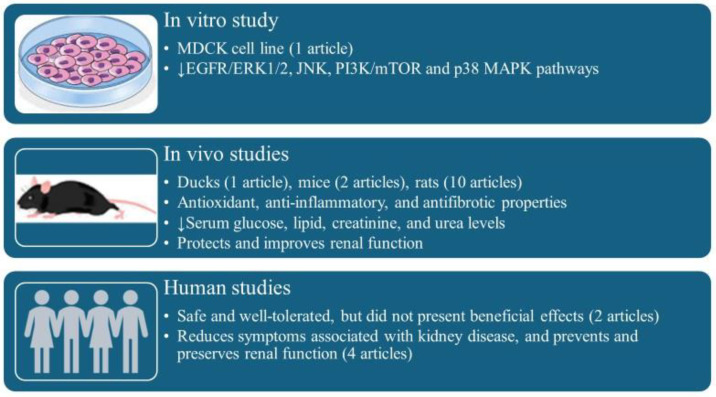
Summary of principal findings from various study types.

**Figure 3 antioxidants-14-01369-f003:**
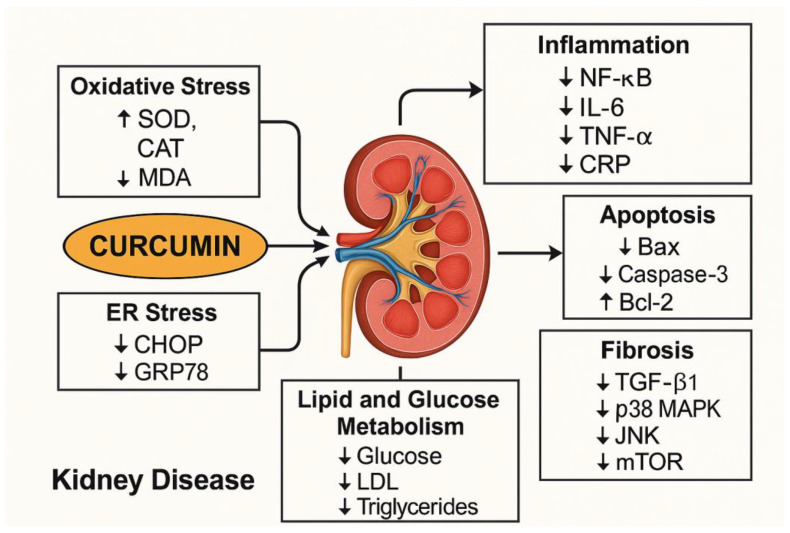
Mechanisms of Actions of Curcumin in Kidney Disease. Curcumin reduces oxidative stress, inflammation, apoptosis, fibrosis, endoplasmic reticulum (ER) stress, and lipid and glucose metabolism. SOD: superoxide dismutase; CAT: catalase; MDA: malondialdehyde; NF-κB: nuclear factor-kappa B; IL-6: interleukin-6; TNF-α: tumor necrosis factor-alpha; CRP: C-reactive protein; TGF-β1: transforming growth factor-beta 1; p38 MAPK: p38 mitogen-activated protein kinase; JNK: c-Jun N-terminal kinase; mTOR: mammalian target of rapamycin; CHOP: C/EBP homologous protein; GRP78: glucose-regulated protein 78. ↑ Increase/Promote; ↓ Decrease/Reduce.

**Table 1 antioxidants-14-01369-t001:** Inclusion and exclusion criteria.

Inclusion Criteria	Exclusion Criteria
Original articles in English	Other types of articles or languages
Curcumin in models of kidney disease	Different effects or models
Drug interactions	Only curcumin treatment
Humans or animals studies	Only in vitro studies

**Table 2 antioxidants-14-01369-t002:** Characteristics of articles included.

Curcumin Alone or in Combination	Disease Condition	Model	Mechanism of Action	Clinical Benefit	Ref.
**In vitro and in vivo studies**					
Curcumin & Ginkgolide B	Autosomal dominant polycystic kidney disease (ADPKD)	In vitro (MDCK) kidney cells In vivo (Pkd1 knockout mouse model)	↓ EGFR/ERK1/2, JNK, PI3K/mTOR ↓ EGFR/ERK1/2, JNK, and p38 MAPK pathways	Reduces kidney cyst size, number, and kidney enlargement	[45]
**Animal studies**					
Curcumin & Rosuvastatin	CKD and related lipid disorders	Adenine-induced CKD in Wistar Rat	↑ eGFR ↑ Serum albumin ↓ Triglycerides, ↓VLDL, LDL ↓ Cholesterol Improved atherogenic and coronary risk indexes	Renoprotective and lipid-lowering effects	[46]
Curcumin	Nephrotoxicity	Gentamicin-induced nephrotoxicity in Sprague Dawley rats	↓ MDA ↑ SOD ↓ BUN, ↑ Creatinine clearance ↓ ER Stress markers (GRP78, CHOP, calpain-2, and caspase-12) ↓ Bax, cytochrome c, cleaved caspase-3 ↑ Bcl-2	Exhibits protective effects by mitigating oxidative stress, ER stress, and apoptosis.	[47]
Nanocurcumin combined with insulin	Diabetic Nephropathy	Streptozotocin-induced nephropathy in in Sprague Dawley rats	↓ BUN, creatinine, ↓ bilirubin, ALP. ↑ albumin, globulin ↓ blood glucose ↑ NLRP3, IL-1β, NF-κB, Caspase-3, and MAPK8 mRNA Deactivates P38 MAPK and P53. signaling pathways basement membrane thickening, tubular atrophy, and podocyte cytoskeletal impairment	Ameliorates hyperglycemia and mitigates inflammation and structural kidney damage	[48]
Curcumin	Nephrotoxicity	Cyclosporine A-induced toxicity in Wistar albino rats	↓ BUN, creatinine ↓ MDA, IL-2 levels. ↑ SOD, CAT, GPx. ↓ tissue damage	Improves renal function, reduces oxidative stress, inflammation	[49]
Curcumin	Focal and Segmental Glomerulosclerosis	Adriamycin -induced FSGS in Wistar rats	↓ Serum creatinine, ↓ BUN, triglycerides, ↓urinary protein ↓ TNF-α, MDA. ↑ SOD, GSH ↓ Segmental glomerulosclerosis	Anti-inflammatory and antioxidant properties reducing kidney damage and preserving renal function	[50]
Curcumin	Hepatorenal Toxicity	Arsenic-Induced toxicity in male albino rats	↓ AST, ALT, ALP, bilirubin, urea, and creatinine levels. ↓ LDL, cholesterol, triglyceride levels, ↑ HDL ↓ MDA ↑ SOD, CAT, GPx, GR preserves tissue architecture.	Improves lipid imbalances, and reduces oxidative stress	[51]
Curcumin-Tagged Cilostazol	Diabetic nephropathy	Streptozotocin and nicotinamide-induced nephropathy in Wistar rats	↓ blood glucose ↓ IL-6 ↓ serum creatinine ↓ BUN ↑ serum albumin levels. ↓ cholesterol, ↓ triglycerides ↓ LDL ↑ HDL	Improves glycemic control, renal function, reduces inflammation, and ameliorates lipid profiles	[52]
Curcumin	Nephrotoxicity	Arsenic trioxide -induced nephrotoxicity in ducks	↓ total cholesterol, triglycerides, LDL ↑ HDL ↓ MDA ↑ SOD, CAT, GPx ↓ Bax/Bcl-2, caspase-3, LC3, Beclin-1 ↓ LC3-II/LC3- ↑ p62 ↓ kidney tissue damage	Reduces oxidative stress, apoptosis, and autophagy markers. Ameliorates dyslipidemia	[53]
Nanoparticle-encapsulated curcumin & epigallocatechin gallate	Diabetic nephropathy	Streptozotocin-induced diabetic nephropathy in mice	↓ Serum creatinine, urea, proteinuria ↓ MDA ↑ SOD, CAT, GPx ↓ TNF-α, IL-6 ↓ Glomerular and tubular damage	Antioxidant, anti-inflammatory, and antifibrotic properties	[54]
Tetrahydrocurcumin, polyenylphosphatidylcholine & Losartan	Diabetic nephropathy and Uninephrectomy	Streptozotocin-induced diabetic nephropathy and uninephrectomy in Sprague–Dawley rats.	↓ BP ↓ albuminuria creatinine clearance ↓ Protein kinase C-α ↓ KIM-1 ↓ Type I collagen ↓ fibrosis ↑ CuZnSOD	Improves blood pressure, reduced markers of kidney injury, and mitigated oxidative stress and fibrosis more effectively than losartan alone.	[55]
Ginger & Curcumin	Diabetic Nephropathy	Streptozotocin-induced diabetic nephropathy in Sprague Dawley rats	↓ Serum creatinine ↓BUN ↓ blood glucose ↓ MDA ↓ IL-6, NF-κB ↑ CAT, SOD, GSH Histopathological Improvements: collagen deposition and glycogen accumulation	Antioxidant, anti-inflammatory, antihyperglycemic, and renal protective effect	[56]
Curcumin (traditional Chinese medicine) in combination with medications for preventing and/or treating diabetic nephropathy	Diabetic nephropathy	Rats	↓ Blood sugar ↓ Serum creatinine ↓ Serum urea nitrogen	The curcumin-containing traditional Chinese medicine exhibits protective and therapeutic effects against diabetic nephropathy	[57]
**Human studies**					
Curcuminoids	Post-contrast acute kidney injury (PC-AKI)	CKD patients undergoing elective coronary angiography	↑ eGFR	Reduces the incidence of PC-AKI and better preserve renal function	[58]
Curcuminoids	Contrast-Induced Acute Kidney Injury	CKD patients undergoing elective coronary angiography or percutaneous coronary intervention	No Significant effect	Curcuminoids were safe and well-tolerated	[59]
Nano-curcumin	Chemotherapy nephrotoxicity	Cisplatin-induced nephrotoxicity in cancer patients	No Significant effect	Nano-curcumin was well-tolerated, with no reported adverse effects	[60]
Turmeric & Piperine	End-Stage Renal Disease	Hemodialysis Patients	↓ MDA ↓ CRP ↓ IL-6 ↑ Total antioxidant capacity	Reduces markers of oxidative stress and inflammation	[61]
Curcumin formulated in combination with taurine, docosahexaenoic acid (DHA), and essential vitamins including A, D3, E, K1, C, B1, B2, B6, B12, biotin, pantothenic acid, and folic acid	Diabetic nephropathy	Diabetic nephropathy patients	↓ Dyslipidemia ↓ Hyperkalemia ↓ Anemia	This invention advances the optimization and personalization of therapeutic nutrition and demonstrates corrective effects on bone mineral disorders in patients	[62]
Curcuma, Astragalus membranaceus, Jinyingzi, Chuanxiong, and Bixie	Diabetic nephropathy	Diabetic nephropathy patients	↓ Proteinuria	The pharmaceutical formulation alleviates clinical symptoms such as fatigue, weakness of the lumbar region and knees, poor appetite, dry mouth, edema, sore throat, and thick, greasy tongue coatings and demonstrates therapeutic efficacy in managing kidney deficiency syndrome	[63]

↑ Increase/Promote; ↓ Decrease/Reduce. PC-AKI: post-contrast acute kidney injury; ER: _endoplasmic reticulum; SOD: superoxide dismutase; CAT: catalase; GPx: Glutathione peroxidase; MDA: malondialdehyde; NF-κB: nuclear factor-kappa B; IL-6: interleukin-6; TNF-α: tumor necrosis factor-alpha; CRP: C-reactive protein; p38 MAPK: p38 mitogen-activated protein kinase; CHOP: C/EBP homologous protein; GRP78: glucose-regulated protein 78.

**Table 4 antioxidants-14-01369-t004:** Summary of Curcumin–Drug Interactions in Kidney Disease.

Combination	General Drug Class of Combination Drug	Observed Effects in Kidney Disease Models	Key Interaction Mechanisms/Clinical Implications	Ref.
Turmeric and Piperine	Herbal Supplement (Turmeric/Curcumin), Bioavailability Enhancer (Piperine/Alkaloid)	Piperine significantly enhances curcumin’s bioavailability	Piperine inhibits drug-metabolizing enzymes (CYP, P-gp), amplifying curcumin’s inherent drug interaction profile. Potentially increased systemic exposure to curcumin and other co-administered drugs	[61,87]
Nanoparticle-encapsulated Curcumin and Epigallocatechin Gallate (EGCG)	Polyphenol (Nano-antioxidant platform)	Enhanced bioavailability and synergistic anti-inflammatory/antioxidant effects. Mitigates acute kidney injury (AKI) in mouse models	EGCG as part of nanoparticle enhances delivery and contributes therapeutic effect. Reduces pro-inflammatory, increases anti-inflammatory cytokines	[54,86]
Tetrahydrocurcumin, Polyenylphosphatidylcholine and Losartan	Curcumin Metabolite, Lipid Carrier, Angiotensin II Receptor Blocker (ARB)	Add-on therapy in diabetic nephropathy rats: lowered blood pressure, enhanced antioxidant defenses, reduced kidney injury/fibrosis markers, improved renal function	Optimized delivery of THC with PPC. Renoprotective effects independent of glycemic status. Potential pharmacokinetic interaction with Losartan via CYP	[55]
Curcumin & Ginkgolide B	Terpene Lactone (PAFR antagonist, antiplatelet, antioxidant, anti-inflammatory)	Synergistic inhibition of cyst formation and enlargement in ADPKD models.	Pharmacodynamic synergy for ADPKD. Potential bleeding risk due to additive antiplatelet effects with other anticoagulants/antiplatelets	[45]
Curcumin & Rosuvastatin	HMG-CoA Reductase Inhibitor (statin)	Synergistic renal protection and antilipemic action in CKD rats	Pharmacokinetic enhancement: Curcumin allows for reduced statin dose to mitigate side effects while potentiate benefits	[46]
Nanocurcumin combined with Insulin	Antidiabetic hormone	Alleviates diabetic kidney disease (DKD) by inhibiting P38/P53 signaling axis, independent of glycemic control. Improved organ function and histology	Nanocurcumin formulation improves bioavailability. Targets inflammation and apoptosis pathways directly, complementing insulin’s glucose-lowering effects	[48]
Curcumin-Tagged Cilostazol	Phosphodiesterase-3 (PDE3) Inhibitor (antiplatelet, vasodilator)	Nanodispersion shows reno- and pancreas-protective effects in diabetic nephropathy rats (improved kidney/lipid profiles)	Advanced drug delivery for synergistic effects. Potential bleeding risk due to additive antiplatelet effects and CYP inhibition	[52]
Ginger and Curcumin	Herbal Supplements (Anti-inflammatory, Antioxidant, Antiplatelet, Hypoglycemic)	Combined anti-inflammatory and antioxidant effects; potential for kidney health	Potential additive antiplatelet effects increasing bleeding risk. Potential additive hypoglycemic effects increasing risk of hypoglycemia	[56,88]

## Data Availability

The raw data supporting the conclusions of this article will be made available by the authors on request.

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
