# Peer review of "Curcumin in the Treatment of Kidney Disease: A Systematic Review with a Focus on Drug Interactions"

_antioxidants, 2025, doi:10.3390/antiox14111369_

Round 1
Reviewer 1 Report
The authors present systemic review on role of curcumin in the treatment of kidney diseases. They emphasize further research and prioritize large-scale clinical trials to evaluate the efficacy and safety of curcumin in diverse kidney disease populations. The manuscript is comprehensive and well-written but there are some minor issues that need to be addressed to improve the manuscript.
- The authors have restricted their search to PubMed and Scopus. They should expand their search to clinical trials.gov, google patents etc. Also, the search is limited to 2021-2025. The review should include research articles from last 10 years to include pharmacological studies, toxicity, animal, and clinical trials.
- List all the in vitro, animal studies, and clinical trials under separate sub-headings. Include a summary table for human clinical trials. Make conclusions based on animal studies and human clinical trials.
- The authors claim that review focuses on curcumin and other drug interactions but most of the studies reported are monotherapy. Include a interaction-focused research study.
- Include separate animal and human studies table.
See above
Author Response
Reviewer 1 comments:
The authors present systemic review on role of curcumin in the treatment of kidney diseases. They emphasize further research and prioritize large-scale clinical trials to evaluate the efficacy and safety of curcumin in diverse kidney disease populations. The manuscript is comprehensive and well-written but there are some minor issues that need to be addressed to improve the manuscript.
- The authors have restricted their search to PubMed and Scopus. They should expand their search to clinical trials.gov, google patents etc. Also, the search is limited to 2021-2025. The review should include research articles from last 10 years to include pharmacological studies, toxicity, animal, and clinical trials.
Response: We thank the reviewer for this suggestion. When we search on clinicaltrials.gov, only two (from 12) completed clinical trials between 2021-2025 agree with the inclusion and exclusion criteria. However, these trials do not have results or publications yet:
Impact of Curcumin and Pentoxiphylline on Chronic Kidney Disease Patients (CKD)
ClinicalTrials.gov ID NCT06458465
Sponsor Alexandria University
Information provided by Alexandria University (Responsible Party)
Last Update Posted 2025-05-28
High-protein Oral Supplement With Liposomal Curcumin in Adults Undergoing Hemodialysis.
ClinicalTrials.gov ID NCT06381076
Sponsor University of Florida
Information provided by University of Florida (Responsible Party)
Last Update Posted 2025-08-03
When we searched on Google Patents, only 3 of 313 agreed with the inclusion and exclusion criteria and have results that are now included in this systematic review:
+Reference [57]: Application of pharmaceutical composition in preparation of medicine for preventing and/or treating diabetic nephropathy
CN117959346A
https://patents.google.com/patent/CN117959346A/en?q=(curcumin)&q=(diabetic+nephropathy+renal+kidney+disease)&before=priority:20250101&after=priority:20210101&language=ENGLISH&type=PATENT&page=19
+Reference [62]: Specialized food product of dietary therapeutic nutrition for patients with diabetic nephropathy
RU2777906C1
https://patents.google.com/patent/RU2777906C1/en?q=curcumin+AND+(%22diabetic+nephropathy%22+OR+%22renal+disease%22+OR+%22kidney+disease%22)&before=priority:20250101&after=priority:20210101&page=8
+Reference [63]: Pharmaceutical composition for treating proteinuria caused by diabetic nephropathy and preparation method and application thereof
CN112773877A
https://patents.google.com/patent/CN112773877A/en?q=curcumin&q=diabetic+nephropathy+renal+kidney+disease&before=priority:20250101&after=priority:20210101&language=ENGLISH&type=PATENT&page=20&peid=642570beaad08%3A356%3A8398b474
The 2021–2025 timeframe was selected to focus on the most recent and relevant evidence, capturing current trends in curcumin research while maintaining a focused, coherent, and scientifically rigorous analysis. Moreover, our manuscript aims to actualize the information systematically reviewed by Jie and cols., Bagherniva and cols., and Emami and cols. These authors reviewed the literature published before 2021, so including this information would be redundant:
[42] Z. Jie, M. Chao, A. Jun, S. Wei, M. LiFeng, Effect of Curcumin on Diabetic Kidney Disease: A Systematic Review and Meta‐Analysis of Randomized, Double‐Blind, Placebo‐Controlled Clinical Trials, Evidence‐Based Complementary Alternative Medicine, 2021 (2021) 6109406.
https://doi.org/10.1155/2021/6109406
[43] M. Bagherniya, D. Soleimani, M.H. Rouhani, G. Askari, T. Sathyapalan, A. Sahebkar, The use of curcumin for the treatment of renal disorders: a systematic review of randomized controlled trials, Advances in Experimental Medicine and Biology, (2021) 327-343.
https://doi.org/10.1007/978-3-030-56153-6_19
[44] E. Emami, S. Heidari-Soureshjani, C.M.J.A.J.o.P. Sherwin, Anti-inflammatory response to curcumin supplementation in chronic kidney disease and hemodialysis patients: A systematic review and meta-analysis, 12 (2022) 576.
https://doi.org/10.22038/AJP.2022.20049
- List all the in vitro, animal studies, and clinical trials under separate sub-headings. Include a summary table for human clinical trials. Make conclusions based on animal studies and human clinical trials.
Response: We thank the reviewer for this valuable suggestion. We have organized studies under separate subheadings for in vitro, animal, and human studies in Table 2. We have now added a summary of the main findings reported in these different types of studies in a paragraph and figure in the Discussion section.
- The authors claim that review focuses on curcumin and other drug interactions but most of the studies reported are monotherapy. Include a interaction-focused research study.
Response: We thank the reviewer for this comment. While our review includes studies of curcumin administered alone, we also include combination studies. Importantly, our focus on drug interactions is emphasized in the Discussion section 4.4 and Table 4: Summary of Curcumin–Drug Interactions in Kidney Disease, which specifically summarizes studies examining curcumin co-administered with other drugs and their potential synergistic or modulatory effects.
- Include separate animal and human studies table.
Response: Thanks for your comments. Table 2 includes separate animal and human studies, and we have now added a summary of the main findings reported in these studies in a paragraph and figure in the Discussion section.

Reviewer 2 Report
The review describing the effects of curcumin in CKD, along with it's drug interactions is generally well written and organized, based on a thorough literature review, selected according to some rigorous criteria. The tables and the figures are suitable and easy to follow. There are no major criticisms regarding this comprehensive and timely review, except for the fact that some additional information should be added about the effect of curcumin on the gastro-intestinal microbiome, as an important modulator of the beneficial effects of curcumin, including the antioxidative effects.
The antioxidants-3953323 manuscript is a well conceived, written and structured review, providing a synthetic literature analysis of the therapeutic effects of curcumin in chronic kidney diseases.
The review is based on up to date references selected through rigorous criteria by applying the PRISMA methodology. The review extensively describes the beneficial, as well as the side effects of curcumin use in therapy. Although there are many scientific articles and reviews regarding curcumin use in the context of kidney disease, this manuscript also emphasizes in a systematic and consistent manner certain pharmacokinetic aspects regarding its interactions with other drugs or pharmacologic substances, which could limit or preclude its use.
The conclusions are well written and appropriate and synthesize the aspects covered in the body of the text.
Although the manuscript covers the most contemporary aspects regarding curcumin use in CKD, information should also be provided about its effects on the gut microbiome, as an important modulator of the bioavailability and pharmacological effects of curcumin. The specific references regarding this issue should be added as well.
Figures and tables are of good quality and the content is suitable.
English language requires only minor corrections, as specified in my report.
The text is well written and the literature screening and selection methodology is suitable. Figures and tables are well designed and informative. English language is adequate, except for some minor corrections that should be needed (Table 3, 'Chronic curcumin administration', in the 'P-glycoprotein row'; line 318 -curcumin 'exerts', instead of 'performs'...)
The 'Discussion' and 'Conclusions' sections are of sufficient length and synthetic, the limitations of curcumin use in CKD being well-pointed as well.
The iThenticate report gives a similitude of 24%. The authors should try to reformulate/change some of the higher similarity sections in order to keep the similarity below 20%.
Author Response
Reviewer 2 comments:
The review describing the effects of curcumin in CKD, along with it's drug interactions is generally well written and organized, based on a thorough literature review, selected according to some rigorous criteria. The tables and the figures are suitable and easy to follow. There are no major criticisms regarding this comprehensive and timely review, except for the fact that some additional information should be added about the effect of curcumin on the gastro-intestinal microbiome, as an important modulator of the beneficial effects of curcumin, including the antioxidative effects.
The antioxidants-3953323 manuscript is a well conceived, written and structured review, providing a synthetic literature analysis of the therapeutic effects of curcumin in chronic kidney diseases.
The review is based on up to date references selected through rigorous criteria by applying the PRISMA methodology. The review extensively describes the beneficial, as well as the side effects of curcumin use in therapy. Although there are many scientific articles and reviews regarding curcumin use in the context of kidney disease, this manuscript also emphasizes in a systematic and consistent manner certain pharmacokinetic aspects regarding its interactions with other drugs or pharmacologic substances, which could limit or preclude its use.
The conclusions are well written and appropriate and synthesize the aspects covered in the body of the text.
Although the manuscript covers the most contemporary aspects regarding curcumin use in CKD, information should also be provided about its effects on the gut microbiome, as an important modulator of the bioavailability and pharmacological effects of curcumin. The specific references regarding this issue should be added as well.
Response: We thank the reviewer for highlighting the potential role of the gut microbiome. In the present study, our primary focus was on curcumin’s direct molecular interactions (including CYP450 modulation, P-glycoprotein activity, and enzyme binding) as well as blood/tissue-level pharmacodynamic effects. While the gut microbiome may influence curcumin’s metabolism and systemic effects, we have re-checked the 19 research results that agree with the inclusion and exclusion criteria, but none of these articles reported information regarding the gut microbiome. For this reason, we are now including this issue within the Limitations and Challenges section, and acknowledged its potential contribution in the Conclusion section, including its possible role in modulating bioavailability and the gut–kidney axis.
Figures and tables are of good quality and the content is suitable.
English language requires only minor corrections, as specified in my report.
The text is well written and the literature screening and selection methodology is suitable. Figures and tables are well designed and informative. English language is adequate, except for some minor corrections that should be needed (Table 3, 'Chronic curcumin administration', in the 'P-glycoprotein row'; line 318 -curcumin 'exerts', instead of 'performs'...)
Response: We thank the reviewer for the correction. We have changed ‘performs’ to ‘exerts’; and include ´administration´.
The 'Discussion' and 'Conclusions' sections are of sufficient length and synthetic, the limitations of curcumin use in CKD being well-pointed as well.
The iThenticate report gives a similitude of 24%. The authors should try to reformulate/change some of the higher similarity sections in order to keep the similarity below 20%.
Response: We thank your suggestion. Although we do not have access to iThenticate, we analyze our manuscript with StrikePlagiarism with few plagiarism coincidences that we re-write.

Reviewer 3 Report
The present review article by Ebenezer et al. aims to summarize the current evidence and the significance of curcumin for the treatment of kidney disease. For this purpose, the authors conducted a literature search following the PRISMA guidelines. From an initial 417 English-language articles, they identified 16 articles that were deemed suitable for systematic review. Of these, 12 were animal studies and 4 were clinical studies as shown in table 2.
Considering only the studies in humans, two were conducted in CKD patients who underwent angiographic examination with contrast medium. While in Ref. 56 curcumin appeared to preserve renal function to some extent in CKD patients exposed to contrast media (30 patients evaluable in each trial arm), the second study (Ref. 57) failed showed to show any effect at all (54 patients randomly assigned to curcumin, 58 to placebo). A third clinical study (Ref. 58) was unable to demonstrate any effect on cisplatin-induced nephrotoxicity (15 cancer patients on curcumin, 15 on placebo). The fourth study of 43 end-stage renal disease patients (Ref. 59) found reduced markers of oxidant stress and inflammation, such as MDA, CRP, IL-6 with a subsequent increase in total antioxidant capacity. I wonder why another study (Ref. 61) is being discussed secondarily although it was not identified in the initial literature search or failed to meet the quality criteria of the PRISMA guidelines.
Based on the available data, the evidence for clinical efficacy of curcumin in patients with CKD is extremely sparse. I strongly recommend presenting the figures I extracted from the original studies in Table 2 to provide the reader with a better quantitative assessment of the available clinical evidence.
Author Response
Reviewer 3 comments:
The present review article by Ebenezer et al. aims to summarize the current evidence and the significance of curcumin for the treatment of kidney disease. For this purpose, the authors conducted a literature search following the PRISMA guidelines. From an initial 417 English-language articles, they identified 16 articles that were deemed suitable for systematic review. Of these, 12 were animal studies and 4 were clinical studies as shown in table 2.
Considering only the studies in humans, two were conducted in CKD patients who underwent angiographic examination with contrast medium. While in Ref. 56 curcumin appeared to preserve renal function to some extent in CKD patients exposed to contrast media (30 patients evaluable in each trial arm), the second study (Ref. 57) failed showed to show any effect at all (54 patients randomly assigned to curcumin, 58 to placebo). A third clinical study (Ref. 58) was unable to demonstrate any effect on cisplatin-induced nephrotoxicity (15 cancer patients on curcumin, 15 on placebo). The fourth study of 43 end-stage renal disease patients (Ref. 59) found reduced markers of oxidant stress and inflammation, such as MDA, CRP, IL-6 with a subsequent increase in total antioxidant capacity. I wonder why another study (Ref. 61) is being discussed secondarily although it was not identified in the initial literature search or failed to meet the quality criteria of the PRISMA guidelines.
Response: We appreciate your detailed revision. Regarding reference [61] (now reference [65]), it corresponds to a publication of 2011, which was not included in our systematic review that covered publications between 2021 and 2025. However, reference 61/65 (2011) was included for its thematic and mechanistic relevance to curcumin’s nephroprotective effects that provided foundational context, complemented recent findings and supported a more comprehensive discussion
- Khajehdehi, M. Pakfetrat, K. Javidnia, F. Azad, L. Malekmakan, M.H. Nasab, G. Dehghanzadeh, Oral supplementation of turmeric attenuates proteinuria, transforming growth factor-β and interleukin-8 levels in patients with overt type 2 diabetic nephropathy: a randomized, double-blind and placebo-controlled study, Scandinavian Journal of Urology Nephrology, 45 (2011) 365-370.
We systematically reviewed articles published between 2021 and 2025 to update the previously revised data by the authors Jie et al. (2021), Bagherniya et al. (2021), and Emami et al. (2022):
[42] Z. Jie, M. Chao, A. Jun, S. Wei, M. LiFeng, Effect of Curcumin on Diabetic Kidney Disease: A Systematic Review and Meta‐Analysis of Randomized, Double‐Blind, Placebo‐Controlled Clinical Trials, Evidence‐Based Complementary Alternative Medicine, 2021 (2021) 6109406.
https://doi.org/10.1155/2021/6109406
[43] M. Bagherniya, D. Soleimani, M.H. Rouhani, G. Askari, T. Sathyapalan, A. Sahebkar, The use of curcumin for the treatment of renal disorders: a systematic review of randomized controlled trials, Advances in Experimental Medicine and Biology, (2021) 327-343.
https://doi.org/10.1007/978-3-030-56153-6_19
[44] E. Emami, S. Heidari-Soureshjani, C.M.J.A.J.o.P. Sherwin, Anti-inflammatory response to curcumin supplementation in chronic kidney disease and hemodialysis patients: A systematic review and meta-analysis, 12 (2022) 576.
https://doi.org/10.22038/AJP.2022.20049
Based on the available data, the evidence for clinical efficacy of curcumin in patients with CKD is extremely sparse. I strongly recommend presenting the figures I extracted from the original studies in Table 2 to provide the reader with a better quantitative assessment of the available clinical evidence.
Response: We appreciate your comments. Following the suggestion of Reviewer 1, we now review three patents that involve human studies. Additionally, we have included a figure that quantitatively represents the clinical results. We dismiss including original results in Table 2, to avoid copyright issues.

Round 2
Reviewer 1 Report
The authors have addressed all the comments.
The authors have addressed all the comments.
Reviewer 2 Report
The authors have made the recommended modifications to the manuscript and have properly responded to the issues previously raised.
This is a well written and timely review and based on the recommendations made during the first first round of review, the authors have made further improvements. The effects and side effects of curcumin in the context of the interaction with other pharmacological substances are well described and the limitations of this study are also listed in a suitable manner. The figures and the tables are informative and of good quality and the references used are adequate.